# Demographic effects on fruit set in the dioecious shrub Canada buffaloberry (*Shepherdia canadensis*)

Kate M. Johnson[1] and Scott E. Nielsen[2]

[1] Centre for Applied Conservation Research, University of British Columbia, Vancouver, BC, Canada

[2] Department of Renewable Resources, University of Alberta, Edmonton, Alberta, Canada

## ABSTRACT

The effects of pollen limitation on reproductive success in plants have been well-documented using pollen supplementation experiments. However, the role of local demographics in determining pollen limitation, particularly in terms of the additive and interactive effects of pollen availability and competition are not well known. We measured fruit set in the dioecious shrub Canada buffaloberry (*Shepherdia canadensis*) in Central Alberta, Canada to evaluate whether local demographics measured at three spatial scales (25, 50, and 100 $m^2$) affect fruit set in buffaloberry. We test whether density-dependence (population density), pollen donor (measured as male density, distance to nearest male plant and size of nearest male plant), female competitor (measured as female density and distance to nearest female plant), or the combined pollen donor and competitor hypotheses best explain natural variations in fruit set for a population of Canada buffaloberry. Support was highest for the combined pollen donor and competitor hypothesis at an intermediate spatial scale of 50 $m^2$. Proportion fruit set increased with male shrub density (pollen donors) and decreased with female shrub density (pollen competitors), but was more affected by the presence of males than females. This illustrates that access to male shrubs within a 3.99 m radius affects pollen availability, while nearby females compete intra-specifically for pollen.

## INTRODUCTION

Fruit set is commonly limited by pollen availability, particularly in dioecious species due to the isolation of male and female reproductive organs (*Burd, 1994*; *Knight et al., 2005*). Because dioecious plants tend to rely on wind or small, generalist insects for pollination (*Bawa, 1990*; *Armstrong & Irvine, 1989*), their reproductive success is dependent on the distribution and density of sexes within the range of insect pollinators or wind transport (*House, 1992*). Demographic factors have been shown to play a role in determining fruit set through regulation of the quantity and quality of pollen available to females (*House, 1992*; *Kunin, 1993*; *Knight et al., 2005*). Pollen quantity is known to be related to distance to nearest conspecific male plant (*Kay et al., 1984*; *De Jong, Batenburg & Klinkhamer,*

Corresponding author
Scott E. Nielsen, scottn@ualberta.ca

2005; *Wang et al., 2013*), local male density (*House, 1992*) and population sex ratio (*Osunkoya, 1999*). However, the role of female competitors in regulating pollen availability, as well as the additive effects of pollen donation from males and competition from females, has received much less attention.

Buffaloberry is a shade intolerant (*Humbert et al., 2007*), nitrogen-fixing (*Hendrickson & Burgess, 1989*; *McCray-Batzli et al., 2004*; *Rhoades et al., 2008*) dioecious shrub common to disturbed boreal and temperate montane forests of western North America (*Stringer & LaRoi, 1970*; *LaRoi & Hnatiuk, 1980*). Shrubs heights (widths are often similar) range between 0.9 and 3.9 m (*Bormann, 1988*), although in Alberta they are rarely over 2 m. Fruit production in buffaloberry is inversely related to canopy cover (*Hamer, 1996*; *Nielsen et al., 2004*), with inter-annual variation in fruit explained primarily by the previous year's midsummer rainfall suggesting that climate affects the development of flower primordium (*Krebs et al., 2009*). Plants flower early in the spring, among the earliest of plants in the region, shortly after the soil thaws and before the forest canopy leafs out (Fig. 1). Buffaloberry is pollinated nearly entirely by dipterans (97%), the majority of which are in the Syrphidae and Empididae families (*Borkent & Harder, 2007*), with Hymenopterans and Hemipterans also known to pollinate buffaloberry (*Lewis, 1990*). Male buffaloberry flowers offer both pollen and nectar rewards to potential pollinators, while female flowers only produce nectar (*Mosquin, 1971*; *Lewis, 1990*). Insect pollinators of buffaloberry visit each sex at equal rates, possibly due to an inability to discriminate between flowers (*Borkent & Harder, 2007*). Pollinators visit an average of 6 flowers per plant, spend >9 s at each flower, and have a 25% re-visitation rate (*Borkent & Harder, 2007*). This relationship suggests that the reproductive success of buffaloberry is pollen-limited due to a deficiency in pollinator visits.

Here we use buffaloberry as a model species to examine how pollen donor (male) and pollen competitors (female) affect fruit set in a dioecious species. We hypothesize that a male-biased population density should produce higher fruit set for any nearby female shrub due to increased pollen availability (pollen donor hypothesis), while a female-biased population density should increase competition for pollen and thus decrease fruit set for any given female plant (female competitor hypothesis). We also test the pollen donor hypothesis as distance to nearest male plant and the female competitor hypothesis as distance to nearest female plant. These could be considered simple pollen donor and competitor hypotheses as commonly measured in the literature. Pollen donor and female competitor hypotheses are not, however, mutually exclusive. Both should affect fruit set in dioecious species and we compare this combined factor hypothesis with the pollen donor and female competitor hypotheses. We consider the pollen donor and female competitor hypothesis as either an additive effect of pollen donors and female competitors (measured as both distance to nearest plant and as local density) or an interactive effect of male and female-biased population, which we interpret as the sex ratio of the population. We also compare these hypotheses against a null model of equal fruit set regardless of local demography, a simple density-dependent hypothesis based on total population size ignoring local sex-biases, and a male size hypothesis that considers the distance and size of

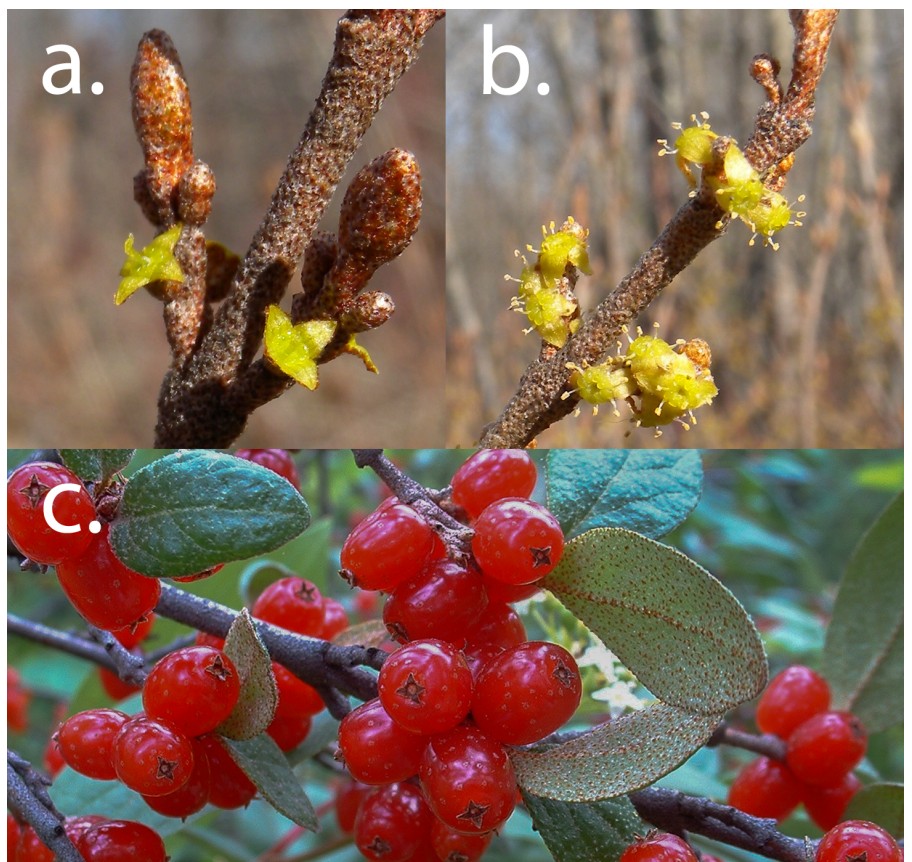

**Figure 1 Canada buffaloberry flowers and fruit.** (A) Pistillate flowers, (B) staminate flowers and (C) ripe fruit of Canada buffaloberry (*Shepherdia canadensis*). Flowers photographed on 6 May 2009 and fruit on 8 July 2004 at Terwilleger Park, Edmonton, Alberta. Male shrubs begin flowering first (sometimes up to 1 week) and are 2–3 times larger than female flowers. Photographs by S Nielsen.

the nearest pollen source. We examine these hypotheses by measuring fruit set for a natural population of buffaloberry in Central Alberta, Canada.

## MATERIALS AND METHODS

Sixty buffaloberry shrubs were randomly selected in Terwillegar Park in Edmonton, Alberta, marked with a double-faced aluminum tag wired to one stem at the base of the shrub, and monitored for flowering and fruit set between 8 May and 22 June 2012. Terwillegar Park is a 174 hectare natural area located along the North Saskatchewan River in the southwest part of Edmonton (53.48071°N, 113.60785°W). The middle of the park is an open off-leash dog area that is surrounded by natural vegetation with minimum management (*City of Edmonton, 2009*). Buffaloberry shrubs in the area are common along forest edges and in semi-open deciduous forests of balsam popular (*Populus balsamifera*) and trembling aspen (*Populus tremuloides*). Total population size in the park is likely >5,000 shrubs. Plants were selected to be representative of the conditions of the study area, ranging from shrubs occurring within low density open habitat to shrubs occurring in more dense edge and forested habitat. In any one location a female plant was randomly

**Table 1** Range and mean number of buffaloberry individuals within each scale of measurement.

| | 25 m² | | | 50 m² | | | 100 m² | | |
|---|---|---|---|---|---|---|---|---|---|
| | ♂ + ♀ | ♀ | ♂ | ♂ + ♀ | ♀ | ♂ | ♂ + ♀ | ♀ | ♂ |
| Range | 0–13 | 0–8 | 0–7 | 0–24 | 0–15 | 0–13 | 0–34 | 0–20 | 0–17 |
| Mean | 3.3 | 1.7 | 1.60 | 6.22 | 3.43 | 2.78 | 10.42 | 5.58 | 4.83 |
| (SE) | (0.44) | (0.26) | (0.23) | (0.78) | (0.46) | (0.39) | (1.20) | (0.68) | (0.58) |

**Notes.**
♀, females; ♂, males; ♀ + ♂, all reproductive plants.

selected by throwing a stake and then selecting the nearest female shrub. Focal females were located a minimum of 11.28 m away from each other to ensure there was no overlap of neighboring shrubs at the largest spatial scale measured (5.64 m radius from focal plant). Shrub density (100 m²) ranged from 0 to 34 reproductive adults around the focal plant, with a mean of 10.42 (±1.20). Our sample population was marginally female-biased at all spatial scales measured but not significantly different (Table 1).

We used a natural experiment to test pollen donor and female competition hypotheses by examining variation in fruit set among shrubs within an open pollination system following other studies of fruit set in dioecious plants (*Armstrong & Irvine, 1989*; *House, 1992*; *Wang et al., 2013*). Although experimental hand pollination experiments (pollinator restriction [bagging] would be unnecessary since it is a dioecious species) could be used to address pollen limitations, we were interested here in examining how local demographic effects influenced fruit set within the same population and year of fruiting.

Due to the large number of potential flowers present on an individual shrub (many 1000 s), a subsample of flowers was counted to measure initial flower production and thereafter fruit set (e.g., *Bowers, 2009*; *Khanizadeh et al., 1989*). Specifically, we systematically sampled from each of the 60 focal female shrubs four branch segments approximately 30 cm in length by randomly selecting one branch from each the four cardinal directions. Sampled branch segment were marked with a Sharpie® pen by encircling the branch stem 30 cm from the tip of the branch with a 'permanent mark'. The number of flowers on each branch segment was counted twice. First between 8 and 10 May and recounted again between 23 and 24 May to ensure full flower counts since phenology of shrubs varied slightly and due to later than normal spring conditions. Because male buffaloberry flowers earlier and longer it generally overlaps with all female flowering. Maximum number of flowers observed among either of the two counts was used as the total number of flowers per sampled branch to ensure flowers were counted during the period of overlap. Because fruit ripening begins here in early July, we visited all shrubs between 22 and 28 of June as the color of fruit began to change color allowing easier counts of the number of fruit per marked branch. Fruit set for each shrub was defined as the proportion of total flowers with fruit based on the number of fruit to flowers counted across all four branch segments (initial analyses revealed no differences among branch orientations). Based on general observations of fruit production in the prior five years, fruit abundance in Terwillegar Park in 2012 appeared to be average (S Nielsen, pers. obs., 2012).

We quantified population demography of local buffaloberry populations around each marked female shrub at three spatial scales: $25 \, m^2$ (2.82 m radius), $50 \, m^2$ (3.99 m radius), and $100 \, m^2$ (5.64 m radius). Prior evidence indicates that the insect pollinators of buffaloberry are short-distance fliers (*Borkent & Harder, 2007*), but because we were unsure of the range within which pollinators are most active, we bracketed sampling across three scales (the moderate scale appears to be the most predictive suggesting that the scale selected was representative of the scale of pollination effects). Distances to all neighboring shrubs (by sex) were measured using a Haglöf DME 201 Cruiser (Långsele, Sweden) with the transponder centered on the marked plant and the electronic receiver held over the center of all other surrounding shrubs to measure distance to the marked female shrub out to a maximum of a 5.64 m radius. In addition to sex-specific densities, distance to nearest male and female shrub was measured as a simple test of the pollen donor and female competitor hypotheses (minimum distance) as this is commonly used in the literature. This included measurements beyond 5.64 m ($100 \, m^2$) in the few cases where shrub density was low enough that no males were present within the largest sampling scale used (5.64 m radius).

Ten *a priori* candidate models were defined for each spatial scale ($25 \, m^2$, $50 \, m^2$, and $100 \, m^2$) based on the following hypotheses (Table 2): (0) *null model* of equal (mean) fruit set among plants (.); (1) *simple pollen donor hypothesis* measured as distance to nearest male shrub or source of pollen ($-\male_{dist}$); (1a) *nearest male and size of nearest male hypothesis* measured as the distance and size of the nearest pollen source ($-\male_{dist} +\male_{size}$); (2) *nearest male and female competitor hypothesis* measured as the distance to nearest male shrub and female shrub density ($-\male_{dist} -\female_D$); (3) *simple pollen competitor hypothesis* measured as distance to nearest female shrub or female competitor ($+\female_{dist}$); (4) *nearest female and pollen donor hypothesis* measured as distance to nearest female shrub and male density ($+\female_{dist} +\male_D$) (5) *density-dependent hypothesis* measured as total population density of reproductive shrubs ($+D$); (6) *pollen donor hypothesis* measured as male shrub density ($+\male_D$); (7) *female competitor hypothesis* measured as female shrub density ($-\female_D$); (8) *pollen donor and female competitor hypotheses* combined additively ($+\male_D -\female_D$) or (9) *pollen donor and female competitor hypotheses (sex ratio)* combined multiplicatively ($+\male_D * -\female_D$). We used the interaction between density of individual sexes rather than a ratio of males to females to represent sex ratio to avoid reducing our sample size as several plots contained no plants within a spatial scale and these observations would have to be excluded in a sex ratio model (inferences were similar when analyzing a smaller set of the data using a male bias variable (male–female shrub density) as a predictor). We predicted the direction of the response in fruit set for each hypothesis as indicated by the − or + symbols representing negative or positive effects on fruit set respectively. Specific to our hypotheses, we expected fruit set to decrease with distance to nearest male shrub ($-\male_{dist}$), and increase with the size of the nearest male ($+\male_{size}$) since that is the source of pollen. Likewise fruit set was expected to increase with population density ($+D$), and especially for male density ($+\male_D$), since again this would be the source of pollen. Conversely, we expected fruit set to increase with distance to nearest female shrub ($+\female_{dist}$)

**Table 2 List of candidate models (hypotheses) predicting fruit set in buffaloberry based on demographic factors and scale of measurement.**

| ID | Hypothesis | Scale | Model |
|---|---|---|---|
| 0 | Null (mean fruit set) | N.A. | . |
| 1 | Nearest male (simple pollen donor) | N.A. | $-\male_{dist}$ |
| 1a | Nearest male & size of nearest male | N.A. | $-\male_{dist} +\male_{size}$ |
| 2a | Nearest male & female competitor | 25 | $-\male_{dist} -\female D_{25}$ |
| 2b | Nearest male & female competitor | 50 | $-\male_{dist} -\female D_{50}$ |
| 2c | Nearest male & female competitor | 100 | $-\male_{dist} -\female D_{100}$ |
| 3 | Nearest female (simple pollen competitor) | N.A. | $+\female_{dist}$ |
| 4a | Nearest female & pollen donor | 25 | $+\female_{dist} +\male D_{25}$ |
| 4b | Nearest female & pollen donor | 50 | $+\female_{dist} +\male D_{50}$ |
| 4c | Nearest female & pollen donor | 100 | $+\female_{dist} +\male D_{100}$ |
| 5a | Density dependence | 25 | $+D_{25}$ |
| 5b | Density dependence | 50 | $+D_{50}$ |
| 5c | Density dependence | 100 | $+D_{100}$ |
| 6a | Pollen donor | 25 | $+\male D_{25}$ |
| 6b | Pollen donor | 50 | $+\male D_{50}$ |
| 6c | Pollen donor | 100 | $+\male D_{100}$ |
| 7a | Female competitor | 25 | $-\female D_{25}$ |
| 7b | Female competitor | 50 | $-\female D_{50}$ |
| 7c | Female competitor | 100 | $-\female D_{100}$ |
| 8a | Pollen donor & competitor | 25 | $+\male D_{25} -\female D_{25}$ |
| 8b | Pollen donor & competitor | 50 | $+\male D_{50} -\female D_{50}$ |
| 8c | Pollen donor & competitor | 100 | $+\male D_{100} -\female D_{100}$ |
| 9a | Pollen donor × competitor (sex ratio) | 25 | $+\male D_{25} * -\female D_{25}$ |
| 9b | Pollen donor × competitor (sex ratio) | 50 | $+\male D_{50} * -\female D_{50}$ |
| 9c | Pollen donor × competitor (sex ratio) | 100 | $+\male D_{100} * -\female D_{100}$ |

and decrease with local female shrub density since they would be competing for pollen ($-\female_D$). We predicted an additive effect of male and female shrub density or distance ($+\male_D/-\male_{dist} - \female_D/+\female_{dist}$) on fruit set with male density and female distance positively related to fruit set and female density and male distance negatively related to fruit set, but not necessarily at the same rate. Finally, we expected an interaction between male and female density ($+\male_D * -\female_D$) above what could be predicted by the additive model, indicating the importance of the local population's sex ratio.

To test support for these hypotheses, we modeled proportion fruit set of buffaloberry based on our hypothesized factors using a generalized linear model (GLM) using STATA 12.1 with a beta distribution and logit link (*Ferrari & Cribari-Neto, 2004*). Collinearity (Pearson correlations > |0.7|) was checked among variables within each model with no problems found. Total number of flowers per sampled length was multiplied by shrub size to represent total flower production, which is often positively correlated with fruit set (*Osunkoya, 1999*; *Somanathan & Borges, 1999*) and was a significant predictor of fruit set.

This variable was included as a covariate in all models. Initial analyses found little effect of environmental variation (canopy cover, broad habitat class and soil electrical conductivity) and thus were not included. Models were ranked for support using the small sample size corrected Akaike's Information Criterion ($AIC_c$) where smaller $AIC_c$ values indicate more support for the model given the data and models tested (*Burnham & Anderson, 2004*). Model parameters were estimated for the top $AIC_c$-selected model (raw $\beta$ coefficients and predicted total response in proportion fruit set when independent predictor variables where changed from observed minimum to maximum values) with predictions graphed to assist with interpretation.

## RESULTS

The most supported candidate models explaining fruit set in buffaloberry were the pollen donor and competitor hypothesis ($+\male_D -\female_D$) at the 50 m$^2$ scale and the pollen donor hypothesis again at the 50 m$^2$ scale ($+\male_{D_{50}}$) (Akaike weights, $w_i = 0.244$ and $0.089$ respectively; Table 3) thus supporting both the pollen donor and female competitor hypotheses. These models were followed by the pollen donor and competitor hypothesis at the 100 m$^2$ scale ($+\male_{D_{100}} -\female_{D_{100}}$) ($w_i = 0.083$; Table 3) and the sex ratio hypothesis at the 50 m$^2$ scale ($+\male_{D_{50}} * -\female_{D_{50}}$) ($w_i = 0.080$; Table 3). The null model (.) of equal fruit set among shrubs, regardless of local demography, was 4.28 times less supported (evidence ratio of Akaike weights, $w_i$) than our top $AIC_c$ model ($\Delta AIC_c = 2.9$). All other models were less supported than the null model ($\Delta AIC_c = 3.5$–$6.5$), but still plausible. This included the simple pollen donor hypothesis that was measured as distance to nearest male plant ($\Delta AIC_c = 3.5$), suggesting that density is a better indicator of available pollen than distance to nearest male. A simple density-dependence model ($+D$) measuring local shrub density also had less support than the null model, illustrating the importance of sex-specific demography and thus opposite effects of sexes on fruit set. The female competitor hypothesis ($-\female_D$) alone had much less support ($\Delta AIC_c = 5.1$ at 100 m$^2$ scale; Table 3), despite being present as a variable in the top supported model which included male density ($+\male_D -\female_D$). Nearest female and pollen donor hypothesis ($+\female_{dist} +\male_D$), as well as the nearest male and female competitor hypothesis ($-\male_{dist} -\female_D$), both had low support indicating that density is still a better predictor of fruit set than distance and density combined. Indeed, the nearest female and pollen donor hypothesis at the 100 m$^2$ scale was the least supported model tested ($\Delta AIC_c = 6.5$; Table 3). Nearest male and size of nearest male ($-\male_{dist} +\male_{size}$) was the second least supported hypothesis ($\Delta AIC_c = 5.9$; Table 3), suggesting that the size of the nearest male does not make the simple pollen donor hypothesis a better predictor of fruit set. When considering the spatial scale at which fruit set was most affected by surrounding shrubs, the 50 m$^2$ scale (3.99 m radius) was consistently more supported than the other two spatial scales tested.

Using the top supported model representing the pollen donor and female competitor hypothesis ($+\male_D -\female_D 50$ m$^2$), proportion fruit set increased by 0.352 (SE $= 0.123$) when male shrub density increased from its minimum (0 shrubs) to maximum (13 shrubs) value ($\Delta$ Min to Max; Table 4). This supports the pollen donor hypothesis where access to male

Table 3 **Ranking of support among candidate models using Akaike's Information Criteria adjusted for small sample size (AIC$_c$).** Hypothesis, model ID, scale (m$^2$), model structure, parameter number ($K$), change in AICc and Akaike weights ($w_i$) are provided. The line within the table separates models ranked higher than the null hypothesis (mean fruit set) from those ranked lower and are thus considered unrepresentative.

| ID | Hypothesis | Scale | Model | K | AIC$_c$ | Δ AIC$_c$ | $w_i$ |
|---|---|---|---|---|---|---|---|
| 8b | Pollen donor & competitor | 50 | $+♂D_{50} -♀D_{50}$ | 4 | −51.7 | 0 | 0.244 |
| 6b | Pollen donor | 50 | $+♂D_{50}$ | 3 | −49.7 | 2 | 0.089 |
| 8c | Pollen donor & competitor | 100 | $+♂D_{100} -♀D_{100}$ | 4 | −49.6 | 2.2 | 0.083 |
| 9b | Pollen donor × competitor (sex ratio) | 50 | $+♂D_{50} * -♀D_{50}$ | 5 | −49.5 | 2.2 | 0.08 |
| 0 | Null | N.A. | . | 2 | −48.8 | 2.9 | 0.057 |
| 1 | Nearest male (simple pollen donor) | N.A. | $-♂_{dist}$ | 3 | −48.2 | 3.5 | 0.043 |
| 6a | Pollen donor | 25 | $+♂D_{25}$ | 3 | −48.2 | 3.5 | 0.043 |
| 8a | Pollen donor & competitor | 25 | $+♂D_{25} -♀D_{25}$ | 4 | −47.9 | 3.8 | 0.036 |
| 6c | Pollen donor | 100 | $+♂D_{100}$ | 3 | −47.6 | 4.1 | 0.031 |
| 4b | Nearest female & pollen donor | 50 | $+♀_{dist} +♂D_{50}$ | 4 | −47.4 | 4.3 | 0.029 |
| 9c | Pollen donor × competitor (sex ratio) | 100 | $+♂D_{100} * -♀D_{100}$ | 5 | −47.3 | 4.5 | 0.026 |
| 5b | Density dependence | 50 | $+D_{50}$ | 3 | −47.1 | 4.7 | 0.024 |
| 5a | Density dependence | 25 | $+D_{25}$ | 3 | −46.8 | 4.9 | 0.021 |
| 2c | Nearest male & female competitor | 100 | $-♂_{dist} -♀D_{100}$ | 4 | −46.6 | 5.1 | 0.019 |
| 7c | Female competitor | 100 | $-♀D_{100}$ | 3 | −46.6 | 5.1 | 0.019 |
| 5c | Density dependence | 100 | $+D_{100}$ | 3 | −46.6 | 5.1 | 0.019 |
| 3 | Nearest female (simple pollen competitor) | N.A. | $+♀_{dist}$ | 3 | −46.6 | 5.1 | 0.019 |
| 7b | Female competitor | 50 | $-♀D_{50}$ | 3 | −46.5 | 5.2 | 0.018 |
| 7a | Female competitor | 25 | $+♂D_{25}$ | 3 | −46.5 | 5.2 | 0.018 |
| 2b | Nearest male & female competitor | 50 | $-♂_{dist} -♀D_{50}$ | 4 | −46.4 | 5.3 | 0.017 |
| 2a | Nearest male & female competitor | 25 | $-♂_{dist} -♀D_{25}$ | 4 | −46.2 | 5.5 | 0.015 |
| 4a | Nearest female & pollen donor | 25 | $+♀_{dist} +♂D_{25}$ | 4 | −45.9 | 5.8 | 0.013 |
| 9a | Pollen donor × competitor (sex ratio) | 25 | $+♂D_{25} * -♀D_{25}$ | 5 | −45.9 | 5.8 | 0.013 |
| 1a | Nearest male & size of nearest male | N.A. | $-♂_{dist} +♂_{size}$ | 4 | −45.9 | 5.9 | 0.013 |
| 4c | Nearest female & pollen donor | 100 | $+♀_{dist} +♂D_{100}$ | 4 | −45.2 | 6.5 | 0.01 |

shrubs affects pollen availability. Female shrub density, on the other hand, was inversely related to proportion fruit set with proportion fruit set decreasing by 0.221 (SE = 0.080) when female shrub density increased from its minimum (0 shrubs) to maximum (15 shubs) value (Table 4) thus also supporting the pollen competitor hypothesis where females compete intra-specifically for pollen. This negative effect on fruit set was, however, evident only after considering pollen donor effects of male shrub density, since there was little support for this effect alone ($w_i = 0.019$ at 100 m$^2$ scale; Table 3). Fruit set was also marginally more affected by the presence of males (pollen donor) than females (pollen competitor) with the highest fruit set occurring when sex bias was skewed heavily towards males (Fig. 2). Across the range of the total flower index (Δ Min to Max), proportion fruit set decreased by 0.283 (SE = 0.065) units (Table 4).

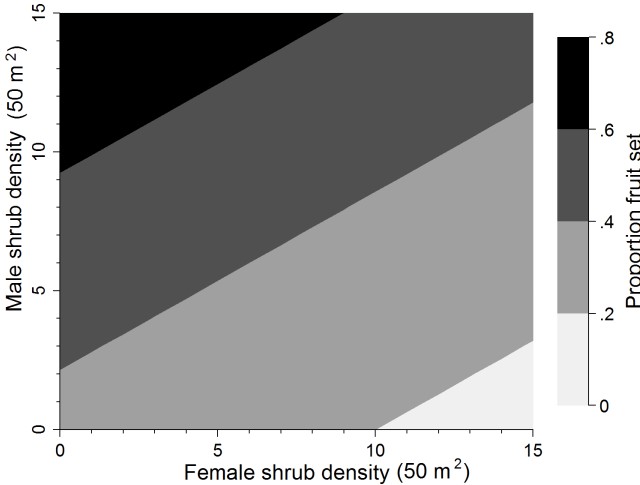

**Figure 2 Sex bias of the most supported model.** Proportion fruit set predicted for buffaloberry based on female and male shrub density.

**Table 4 Model parameters and predicted total response in proportion fruit set for the top AIC$_c$ selected model.** Model coefficients ($\beta$) for variables included in the most supported (AIC$_c$) candidate model describing fruit set in buffaloberry as the pollen donor and female competitor hypothesis (50 m$^2$ scale). Change in the predicted dependent variable when the explanatory variable changes from its minimum to maximum value (while holding other factors at their mean value) is reported as Δ Min to Max. Note that flower index was used to represent total flower production (total sub-sample of flowers on shrub × shrub size) and was included as a covariate in all models ($\beta$ reported here as 1,000 times its real value given its absolute effect per flower index is small).

| | | | 95% Conf. Interval | | Δ Min to Max | |
| --- | --- | --- | --- | --- | --- | --- |
| **Variable** | **$\beta$** | **SE** | **Lower** | **Upper** | **Coef.** | **SE** |
| ♂$D_{50}$ | 0.114 | 0.041 | 0.035 | 0.194 | 0.352 | 0.123 |
| ♀$D_{50}$ | −0.073 | 0.035 | −0.143 | −0.004 | −0.221 | 0.080 |
| Flower index | −0.179 | 0.076 | −0.327 | −0.031 | −0.283 | 0.065 |
| Constant | 0.510 | 0.152 | −0.808 | −0.213 | | |

## DISCUSSION

Although a number of studies have demonstrated a negative effect of plant isolation on fruit set, especially for dioecious plants (*Kay et al., 1984*; *House, 1992*; *Steffan-Dewenter & Tscharntke, 1999*; *De Jong, Batenburg & Klinkhamer, 2005*; *Wang et al., 2013*), pollen limitation is not considered in terms of the additive or interactive effects of pollen availability (pollen donor) and competition (surrounding female shrubs). We found that density of male and female shrubs at both 50 m$^2$ and 100 m$^2$ predicted fruit set in buffaloberry better than nearest neighbor measures. Similar to other studies, local male density was significantly more related to fruit set than distance to nearest male (*House, 1992*). Our results may be related to the foraging habits of buffaloberry pollinators, which visit an average of 6 flowers per plant, spend a relatively long time at each flower and revisit

flowers frequently, indicating that pollinators are not highly mobile and tend to forage within a small area (*Borkent & Harder, 2007*), making a high concentration of local pollen important.

Local male density was most predictive of fruit set when considered in terms of the additive effect of increased pollen donors and decreased pollen competitors. Our support for the female competitor hypothesis contrasts with the findings of *Wang et al. (2013)* who found no significant effect of female competition on fruit set in the dioecious tree *Rhamnus davurica*. We show that female density alone is not a good predictor of fruit set unless considered in conjunction with male density, indicating that future assessments of pollen limitation should consider these factors in terms of their additive or multiplicative effects. Distance to nearest female is also not predictive of fruit set, even when considered with male density, suggesting that density is a better measure of female competition than distance to nearest female. Competition for pollen at high female densities limits the quantity of pollen available to any given female. Because buffaloberry pollinators visit both sexes at equal rates (*Borkent & Harder, 2007*), pollinators are more likely to have visited a female previously and be carrying less pollen in a population with high female density. In addition to facilitating higher fruit set, females occurring within male-biased populations may experience increased long-term fitness. Females with access to a wider choice of mates could produce a surplus of embryos, which would enable selective abortion of lower quality seeds (*Melser & Klinkhamer, 2001*). The reproductive advantages attributed to females occurring within male-biased populations may be necessary to compensate for the greater reproductive costs incurred by females that attract seed dispersers with fleshy fruits. Indeed male-biased sex ratios are common in other long-lived dioecious species with biotic seed dispersal and fleshy fruit (*Field, Pickup & Barrett, 2012*). The marginally higher female-bias of our sample population suggests that reproductive success is at least partly limited by pollen quantity.

All demographic factors were most predictive of fruit set at a scale around focal female shrubs of 50 m$^2$, likely due to the combined effects of plant distribution and pollinator activity. Similar studies of dioecious species have documented a threshold of isolation below which fruit set is not limited by insufficient pollinator visits (*Kay et al., 1984*; *De Jong, Batenburg & Klinkhamer, 2005*). It is likely that the high flight costs of generalist pollinators confine pollinator activities to a small area and discourage travel between patches (*Klinkhammer, de Jong & Linnebank, 2001*). The high number of flowers visited per plant and rate of re-visits (25%, *Borkent & Harder, 2007*) indicates pollinator reluctance to leave a patch once they have begun foraging. A lack of support for the smaller 25 m$^2$ scale suggests that plant distribution is also important. At this smaller scale there may not have been sufficient males to provide the benefits of increased pollen availability and the benefit of pollen donors was therefore not detected.

In contrast with similar studies (*Osunkoya, 1999*; *Somanathan & Borges, 1999*), we found a negative relationship between our index representing total flower production and fruit set. Undiscriminating pollinators may cause large females with many flowers to be at a disadvantage, resulting in more severe pollen limitation and lower reproductive success.

The lack of support for the density dependent hypothesis indicates that as well as not biasing visits based on sex or floral productivity, pollinators also do not prefer patches with a higher density of plants. This suggests that the Dipteran pollinators of buffaloberry are opportunistic, and given similar rates of pollinator visitation to males and females, and low and high density patches, females located within male biased populations are least likely to be pollen-limited and will therefore experience higher fruit set.

## CONCLUSION

A male-biased population of buffaloberry surrounding a female shrub (within 3.99 m radius; 50 m$^2$) exhibits higher fruit set, supporting both the pollen donor and female competitor hypotheses. Although fruit set in buffaloberry was influenced by both male (positively) and female (negatively) shrub density, local male density had a stronger effect on fruit set. This study demonstrates that local demographics affect fruit set through the additive effects of pollen donors and competitors. More research is needed to understand factors affecting flower production and pollinators of buffaloberry.

## ACKNOWLEDGEMENTS

We thank Andrew Braid and Tyler Bateman for their assistance during fieldwork.

### Funding

This work was funded by the Natural Sciences and Engineering Research Council of Canada. The funders had no role in study design, data collection and analysis, decision to publish, or preparation of the manuscript.

### Grant Disclosures

The following grant information was disclosed by the authors:
Natural Sciences and Engineering Research Council of Canada.

### Competing Interests

The authors declare there are no competing interests.

### Author Contributions

- Kate M. Johnson performed the experiments, analyzed the data, wrote the paper, prepared figures and/or tables.
- Scott E. Nielsen conceived and designed the experiments, analyzed the data, contributed reagents/materials/analysis tools, prepared figures and/or tables, reviewed drafts and revised the paper.

### Supplemental Information

Supplemental information for this article can be found online at http://dx.doi.org/10.7717/peerj.526#supplemental-information.

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
