# Peer review of "Demographic effects on fruit set in the dioecious shrub Canada buffaloberry (Shepherdia canadensis)"

_PeerJ, doi:10.7717/peerj.526_

## Round 0.1 · original submission · Major Revisions

All of the reviewers felt the manuscript was sufficiently sound for publication in PeerJ, but nonetheless raised a few concerns about the analysis. In particular, I would urge you to consider including analysis of frequency and sex ratio (or a discussion of why it is not appropriate to do so). The reviewers also raise several points about clarity and interpretation that should be considered as well.

Reviewer 1 ·

Basic reporting

Introduction does a good job of introducing the topics to be analyzed, however I think it may be improved if it included a brief review of the factors that affect fruit set on dioecious species.

Some references are not listed in the bibliography section: Wang, et al. 2013, House 1992, please make sure that all references are properly listed.

Experimental design

The experimental design is able to answer the questions posed by the authors. However I have a few questions that need to be addressed in order to clarify some issues. Statistical analyses need to consider the binomial nature of the response variable. Specifically:
Regarding the 60 focal females and their corresponding neighbourhoods, how often did these neighbourhoods overlap. If they did overlap, wouldn’t this have an effect on the density effect of females?

A description of the population should be included: density, spatial distribution, an approximation of population size. Were plants collected along a trail or path or

The authors mention that flowering phenology varied among individuals, and similar to other dioecious species, male flower earlier. If flowering synchrony among males and females varies, density estimates are likely to be overestimated. A brief description of flowering phenology and the synchronicity between male and female flowers should be included.

How was floral productivity of each adult taken into account? Male and female density estimates assume floral production is comparable among all individuals. It has been shown (Somanathan & Borges 2000) that floral productivity is a significant component of fruit set in dioecious species.

On line 97, not sure what you mean by “counted on the different dates across all four branches…”. I thought that flowers were counted on two accounts and the largest number was taken. Were fruits counted on multiple dates ? If so, how were these counts summarized?

I am a bit unclear on how density and sex ratios are taken into account in your models. The effect of males on fruit set is very likely to be different if these males are found in a neighborhood with three females or with 30 females. Pollen availability is clearly going to be impacted. To me this should be modeled as an interaction between male and female densities, or include sex ratio in the models. An additive effect of male and female density does not describe the interaction between these two components. The effect of sex ratio should be included. If it is going to be ignored then a better explanation than the one provided on lines 117-118 should be provided.
Density should be included as a covariable when looking at distance to the nearest male.

Lines 130-131. Fruit set is a binomial variable, therefore a GLM with a binomial family is better suited for this sort of data. This analysis also controls for the number of flowers used to estimate fruit set, therefore it may have a significant impact on your conclusions.

AICc was used to select the best model, however based on Akaike’s weights the distance model appears to be a valid model. It shouldn’t be dismissed entirely and perhaps discussed as a likely model. Other studies in dioecious species have found a significant effect of male distance on fruit set, therefore your confidence in rejecting a model that includes this factor should be discussed in more detail.

Validity of the findings

your results (Figure 2) suggest that sex ratio is an important factor affecting fruit set in this species. As stated above I do not think analyses take sex ratio into account, since male and female densities, even if they are included in an additive model, do not explicitly analyze sex ratio. Additionally, fruit set should have been analyzed as a binomial variable, which could result in different findings. Therefore the validity of the findings depends on a reassessment of the analyses and the resulting conclusions.

Additional comments

Lines 171-180. This paragraph needs to be revised as it seems to present some contradictions. YOu find that male (+) and female (-) densities significantly affect fruit set. House 92 also finds similar results, she finds that local male density does have an effect on female RS. Additionally, your last sentence in this paragraph (line 180) seems to contradict your findings, you find that distance is not as important as local density of males and females (sex ratio).

Line 187. Why didn’t you include a multiplicative effect (i.e., interaction). This would effectively take sex ratio into the model.

Discussion refers repeatedly to male or female biased populations, however as previously stated you have chosen not to include sex ratio in your analysis. This appears to be a disconnect between your analysis and your conclusions.

·

Basic reporting

Overall I would say the article clearly meets PeerJ standards. I noticed a few minor instances where the clarity of the text could be improved. Should the authors wish to improve clarity, I have suggestions below:

Abstract/various: The 3.99 meter radius area and 50m-square area describe the same thing, and are used interchangeably. This is not immediately obvious from the first use of the numbers (without the reader doing math). I would suggest saving the reader from math.

Lines 25-26: The reader is confused by an abrupt change in topic from heterospecific pollen to possible advantages of dioecy. I think the authors are intending to contrast the costs and benefits of dioecy? Rephrasing would clarify.

Lines 197-198: It is unclear if the authors mean to say that buffaloberry is indeed male-biased overall in its populations. It would be relevant to clarify whether the species is or is not male-biased overall.

Lines 219-220: No discussion has yet been made about factors affecting flower production, this is an odd concept to end on for a paper that has little discussion on flower production. However, it is an interesting and highly relevant topic, perhaps the authors would expand?

Experimental design

I think the experimental design was overall rigorous and of appropriate technical standard to be deserving of publication. I have two points, however that I believe should be addressed.

Was anything done to control for natural environmental variation between focal female sites? i.e. Some places might be cooler or shadier than others and affect pollinator activity/fruit set independently of pollen donors and competitors. If something was done, or if data are available on this, I recommend that the authors include this. If nothing was done, I recommend that the authors consider commenting on any influence of environmentally variability, or lack thereof, on pollinators/fruits.

Why weren't male distance and female density tested as a separate model? Given the comments on lines 175-180, it seems that this would be an appropriate model to test. It seems the authors expect male distance might be more important than male density, so why not see if a model with nearest male distance and local female density outperforms the current best model?

Validity of the findings

No comments.

Additional comments

This paper was generally clear and well-written, and a solid analysis of a useful experiment. In my opinion, this is good quality science and the paper should be accepted for publication in PeerJ.

However, I do have two points that I think should be addressed for final publication: that of commenting on how environmental variability between sites would or would not affect pollinators/fruit set, and that of testing one additional linear model which seems relevant to the hypotheses of the authors.

Reviewer 3 ·

Basic reporting

I found some of the introduction a bit less clear than I would prefer. I specified a couple of examples in my comments to the authors.

The naming of the hypothesis is another area that I would recommend making some changes. It seems confusing to use similar names to refer to different hypothesis.

Experimental design

The experimental design was appropriate, though increasing population size would make the results more general than just to this one population of this one species.

Validity of the findings

My suggestions can be found in my specific comments to the authors. I think adding female or male frequency should be included in this study. Though the finding seems clear, the interpretation needs to be better thought through.

Additional comments

This manuscript describes a study that used a dioecious shrub (Canada buffaloberry) to examine the effect of surrounding male and female density on the seed production of focal females to evaluate several hypotheses regarding pollen availability and pollen competition. It is predicted that higher pollen source (male shrubs) will be beneficial while higher pollen competition (female shrubs) will be detrimental to fruit set of a female plant. Data used to test these hypotheses were collected from one natural populations and the main method used was regression analysis and comparisons among models including different predicting variables.

The questions addressed in this manuscript are important and interesting to plant evolutionary biologists. There were sufficient preliminary results from other studies to suggest that this was an appropriate system for this question. Though I like the fact that this study was carried out in natural population and, therefore, represents a natural phenomenon, the sample size of one population makes it less general. I have a few other comments that I list below:

1. The second sentence in Introduction section is confusing to me. It states: “As a result of having half the number of seed bearers as monecious species, dioecious plants must produce more seeds and disperse those seeds to as many areas as their hermaphroditic counterparts, but with twice the population density (Heilbuth et al. 2001).”
a. First, a minor point: “monoecious” is the more common spelling than “monecious”.
b. This is in the context, I believe, that if dioecious variant is to establish against a hermaphroditic ancestral form, it would somehow need to do better than the hermaphrodites. But, do they have to produce more seeds AND have twice the population density? I am not sure I understand/agree with this statement and I do not think Heiluth et al. (2001) stated this, either.

2. The sentence in lines 25-27 seems a bit misleading to me. I do not think that obligate outcrosser can decrease the accumulation of harmful mutations. In fact, the commonly considered idea of “genetic purging” – that inbreeding (and selfing) can expose deleterious alleles to selection and lead to less genetic load, would counter the argument presented by the authors here.

3. In a few places, I found authors to be specifying that there is an additive effect of pollen availability and pollen competition. Why emphasizing the additive effect, as oppose to non-additive (more specifically, a female frequency) effect? It is often found that neighborhood sex ratio is an important factor determining reproductive success of females. I think it would be worth it to also test the effect of frequency (v.s. density) on reproduction. From what I can tell, the authors mentioned that because in some areas only one sex was found and, hence, sex ratio was not appropriate to use. Why not just use frequency of female (or male) as F/(F+H)? This ratio will range between 0 and 1 and should eliminate the authors’ concern.

4. Related to the last comment, the fact that the (+♂D –♀D) model is the best among ones tested but not either (+♂D ) or (–♀D) individually, suggests that these two factors work together to influence reproductive success of the focal female plants. It does make me think that using frequency might be a good way to tease apart whether it is just the frequency (independent of density) or if both frequency and density are equally important.

5. Line 76: the phrase “a natural experimental design” sounds a bit awkward. I suggest changing it to “a natural experiment”.

6. Is there any justification of using the three distances scales? If so, please add in the manuscript.

7. I assume that the three spatial scales are in the relationship of concentric circles, not concentric rings, correct?

8. What does the line in Table 2 indicate?

9. I found it confusing to use the same names (pollen source and pollen competitor) to refer to different hypotheses (line 111-116), especially during results section on which hypothesis was supported when the same name could be referring to different hypothesis. I suggest that different names be given to different hypotheses to make it clearer.

10. The conclusion on line 200 – because the 50m2 scale turned out to be most predictive, it was concluded that the pollinators were transferring pollen most actively between plants that are within a 3.99m radius of each other. Though this is a direct interpretation of the results, it seems less clear to me what it means biologically. If we think about what pollinators do in natural populations, it seems to me that anything within this spatial range should show the signal of such effect if pollinators are just much more actively within this spatial scale. However, results from the 25m2 scale did not show the same pattern. A more likely mechanism underlying the pattern found in this paper might be due to combined effects of distribution of the plants (within this population) and the pollinator activity. Perhaps, within a smaller scale, there was just not sufficient males to provide that benefit of having pollen donors and, therefore, the signal can only be found at the intermediate scale.

---

## Round 0.2 · accepted · Accept

Many thanks for your careful response to reviewers. I'm happy to report that the manuscript is now acceptable for publication. A few minor comments: I noticed in the manuscript conclusion that "pollen donor" and "pollen competitor" were still used, and there is a typo in the abstract. You will likely want to fix these small issues before uploading a final draft.